# Gelatin Reinforced with CNCs as Nanocomposite Matrix for *Trichoderma harzianum* KUEN 1585 Spores in Seed Coatings

**DOI:** 10.3390/molecules26195755

**Published:** 2021-09-23

**Authors:** Bianca-Ioana Dogaru, Vasile Stoleru, Gabriela Mihalache, Sems Yonsel, Maria-Cristina Popescu

**Affiliations:** 1Laboratory of Physical Chemistry of Polymers, Petru Poni Institute of Macromolecular Chemistry, Romanian Academy, 700487 Iasi, Romania; dogaru.bianca@icmpp.ro; 2Department of Horticulture, “Ion Ionescu de la Brad” Iasi University of Life Sciences, 700490 Iasi, Romania; vstoleru@uaiasi.ro (V.S.); gabriela.mihalache.gm@gmail.com (G.M.); 3Integrated Center of Environmental Science Studies in the North Eastern Region (CERNESIM), Alexandru Ioan Cuza University, 700506 Iasi, Romania; 4ORBA Biokimya San. ve Tic. A.S., Istanbul 34956, Turkey; yonsel@simbiyotek.com

**Keywords:** gelatin, cellulose nanocrystals, bionanocomposite, seed coating, biofertilizer, germination, plant growth

## Abstract

Increasing interest on sustainable agriculture has led to the development of new materials which can be used as seed coating agents. In this study, a new material was developed based on gelatin film reinforced with cellulose nanocrystals (CNC) which was further used as nanocomposite matrix for *Trichoderma harzianum* KUEN 1585 spores. The nanocomposite films were characterized by Fourier transform infrared spectroscopy (FTIR), X-ray diffraction (XRD) and scanning electron microscopy (SEM), showing the formation of new hydrogen bonds between the components with a good compatibility between them. Measurements of water contact angles and tests of water vapor sorption and swelling degree revealed an improvement in the water vapor absorption properties of the films as a result of their reinforcement with CNC. Furthermore, by adding the *Trichoderma harzianum* KUEN 1585 spp. in the seed coating material, the germination percentage, speed of germination and roots length of the corn seeds improved. The polymeric coating did not inhibit the growth of *T. harzianum* KUEN 1585, with this material being a good candidate in modern agriculture.

## 1. Introduction

The economic and environmental future all over the world will profoundly be affected by the development of biodegradable and biocompatible materials derived from renewable resources [1]. In the last decades, to meet the compliance objective, different biopolymers, such as lipids, polysaccharides or proteins [2,3] have been used to synthesize or to produce different polymeric matrices, nano-delivery systems or nanocomposite films due to their biocompatibility, biodegradability and improved mechanical and physical properties. It is known that films based on proteins and polysaccharides have good mechanical properties, but they have high water vapor permeability and are sensitive to moisture, while gelatin-based films have proven to be more resistant and impermeable compared to those produced by polysaccharides [4].

Gelatin is typically obtained from animal bones, skin or ligaments, being a by-product from the meat industry. It is a soluble, flavorless and slightly yellow biopolymer composed of proteins (85–92%), minerals and water, and it is obtained by partial hydrolysis of collagen [5]. Due to its properties, including its highwater binding and film-forming capacity, gelatin is largely used in many industries, like food, cosmetics and pharmaceutics [6], and shows promising results to be used in the agricultural sector. Its films have good mechanical properties, but do not have sufficient water resistance and exhibit poor barrier properties against water vapors [7]. This is an important disadvantage when gelatin is used in high moisture environments because films may disintegrate in contact with water. To optimize the film properties, different nano-sized structures as reinforcing fillers or different plasticizers [8] were studied, thus obtaining nanocomposites with superior properties.

One of the most studied polysaccharide-based nanomaterials in polymer nanocomposites are cellulose nanocrystals (CNC). Generally, CNCs are obtained by acid hydrolysis, degrading the amorphous regions of the cellulose fibers and preserving the crystalline ones. Their dimensions are influenced by the initial source of cellulose, having different widths (5–20 nm) and lengths (100 nm–2 μm) [9]. The main properties which make CNCs popular are their biodegradability, renewability, non-toxicity and abundance. They also exhibit high elastic modulus and high tensile strength [10,11]. In their study, Leite et al. [12] observed that by adding only 0.5 wt% of CNCs led to an increase in tensile strength and Young’s modulus of gelatin. Santos et al. (2014) found that gelatin nanocomposites showed high stiffness and good ductility up to 5 wt% CNCs. Pesticides have been used in agriculture for increasing food productivity since World War II, having a positive impact on pest management. In modern agriculture, to protect germinating seeds and seedlings from insects and soil-borne pathogens, many types of seeds (e.g., corn [13], oilseed rape, cotton, sunflower [14], tomatoes [15], etc.) are coated with active ingredients [16]. However, despite all the advantages, the use of pesticides is avoided nowadays due to the social and environmental damage, and also due to their negative effect on human health [17]. Thereby, a promising alternative is the use of biological and natural agents—biopesticides in a seed-coating formulation [16,18].

Generally, seed coatings consist of an adequate coating material directly applied on seeds, forming a thin and uniform layer. The structural materials used to cover the seeds are classified into binders, fillers, active ingredients, polymer matrix and water [19]. It is important that coating agent can adhere to seeds, and once planted, have a positive impact on germination and seedling vigor. Therefore, biodegradable polymers can be used as coating material because they can serve as nutrients for biocontrol agents [20].

One of the widely studied biocontrol agents is *Trichoderma* spp., with good results on soil or seeds as a growth-promoting agent or applications against several foliar diseases [21].

The aim of this study was to reinforce gelatin films with CNCs through the solvent casting method, and the films’ structural features and properties were evaluated. Furthermore, the feasibility of using gelatin-based corn seed coating incorporated with biofertilizer fungus *Trichoderma harzianum* KUEN 1585 and its impact on germination of corn seeds was tested. The originality of this study exists in the use of CNC as a nanofiller which can provide moisture absorption and retention capability to help seed germination and to enhance the shelf life of microbial fertilizers at ambient conditions. Moreover, the development of microbial formulations requires the use of biocompatible materials.

## 2. Results and Discussions

### 2.1. Film Characterization

#### 2.1.1. Fourier Transform Infrared Spectroscopy

Infrared spectroscopy was used to identify the chemical structure and the interactions established between the pure components during the preparation of the film. Because the spectrum of the CNC partially overlaps with the spectrum of gelatin, to provide evidence of the interactions taking place in the composites the experimental spectra and their derivatives were compared with the calculated ones (using the components’ spectra and the low additivity) [22]. For an immiscible blend, the calculated and the experimental spectra should be the same, while for a miscible blend, differences can occur due to the interactions between the components [23].

FTIR spectra and their derivatives of the composite films and their components are presented in Figure 1. The spectra were separated in two regions: 3750–2700 cm^−1^ assigned to stretching vibration of hydroxyl and methyl/methylene groups and “fingerprint region” between 1800–800 cm^−1^.

From the experimental spectra small differences can be observed, but the second derivative spectra highlight the main differences among the spectra, mostly in the CH stretching vibration region [24]. The experimental and calculated spectra and their second derivatives of the samples in the region 3750–2700 cm^−1^ are presented in Figure 1.

It can be seen that both components have a broad band in the 3750 and 3000 cm^−1^ regions assigned to different stretching vibrations of OH groups, and two bands in the 3000–2700 cm^−1^ spectral region assigned to symmetric and asymmetric stretching vibration of CH_3_ and CH_2_ groups [24].

By analyzing the second derivative spectra of the pure components, the following bands are highlighted. For gelatin, characteristic bands can be observed at 3408 cm^−1^ assigned to of N-H and O-H stretching vibrations (amide A) [25], at 3272 and 3144 cm^−1^ assigned to free N-H groups and to H-bonded N-H groups, and at 2932 cm^−1^ assigned to stretching of C-H and –NH_3_^+^ (amide B) [12]. For cellulose nanocrystals a broad band at 3410 cm^−1^ is observed in the spectrum. This is actually composed of a series of sub-bands corresponding to different intermolecular and intramolecular hydrogen bonds in cellulose I: 3566 cm^−1^ assigned to free OH(6) and OH(2) in cellulose and weakly absorbed water, 3411 cm^−1^ assigned to O(2)H…O(6) intramolecular hydrogen bonds and results in the formation of crystalline regions in cellulose, 3336 cm^−1^ assigned to O(3)H…O(5) intramolecular hydrogen bonds in cellulose, 3275 and 3223 cm^−1^ assigned to O(6)H…O(3) intermolecular hydrogen bonds in cellulose monoclinic Iβ and triclinic Iα phases. In the methyl/methylene region, three bands are observed in the second derivative spectrum of CNC at 2961, 2903 and 2853 cm^−1^ assigned to symmetric stretching vibration of methyl and methylene groups [25].

From the theoretic spectra, all the bands observed in the composite spectra are only a sum of intensities of the component bands, presenting higher intensities and increasing proportionally with the concentration of the components in the blends. Moreover, they present a similar position with the bands found in gelatin (the component with the higher concentration), with the exception being observed for the band from 3337 cm^−1^ which is present only in the CNC spectrum. This later band is increasing in intensity proportionally with the increase in the CNC content.

By comparison, the experimental second derivative spectra show differences as follows: the band from 3337 cm^−1^ (in CNC) (assigned to O(3)H…O (5) intramolecular hydrogen bonds) is shifted to 3332/3334 cm^−1^ in the composite spectra and the intensity does not increase proportionally with the increase in the CNC content (as in the case of calculated spectra); the band from 3275 cm^−1^ (in CNC)/3276 cm^−1^ (in G) (assigned to intermolecular hydrogen bonds) is shifted to a higher wavenumber in the composite spectra (to 3278–3281 cm^−1^) and presents lower intensity when comparing to component spectra and also comparing to the same band from the calculated spectra. Furthermore, the band from 3067 cm^−1^ (in G) (assigned to CNH bending vibration) is shifted to 3072 cm^−1^ in composite spectra and also shows a lower intensity when comparing to the one from the calculated spectra. It is also interesting to note that the band from 2961/2967 cm^−1^ (in CNC/G) (assigned to symmetric stretching vibration of methyl and methylene groups) is not observable in the composite spectra with 10% and 15% CNC, while the band from 2903 cm^−1^ (in CNC) (assigned also to symmetric stretching vibration of methyl and methylene groups) is not visible at all in the composite spectra.

All these differences observed between the calculated and experimental spectra indicate the presence of interactions taking place between the gelatin and CNC via hydrogen bonds.

In the fingerprint region, the characteristic bands for gelatin appear at 1683 cm^−1^ assigned to β-Turn [26], at 1668 cm^−1^ assigned to amide I: C=O stretch/hydrogen bond coupled with COO-, and at 1626 cm^−1^ assigned to amide I (C=O stretching vibration of amide and a small contribution of C-N stretching vibration); the band with a maximum at 1519 cm^−1^ assigned to amide II (C-N stretching vibration and N-H bending vibration), and the band from 1237 cm^−1^ assigned to amide III (C-N stretching vibration coupled to N-H bending vibrations with small contribution from C-C stretching and C=O in plane bending vibration) [27]. The band from 1519 cm^−1^ is a large band which includes both types of vibrations for the α-helical structure (with a maximum at 1550–1540 cm^−1^) and β-sheets structure (at 1525–1520 cm^−1^) [28].

Other important bands in the fingerprint region of the gelatin spectrum are observed at 1447, 1395, 1279, 1197, 1159, 1116, 1081, 1028 and 971 cm^−1^. These are assigned to the CH deformation vibration, OH in plane deformation vibration, CH bending vibration, C-N stretching vibration in amine groups, C-O-C stretching vibration, C-C stretching vibration and C-O stretching vibration, respectively [23,24,25].

In addition, the spectrum of CNC shows some characteristic bands at 1458, 1427, 1369, 1337, 1314 and 1275 cm^−1^ assigned to stretching and deformation vibration of CH groups; at 1203, 1159, 1107 and 1055 cm^−1^ assigned to stretching and deformation vibration of C–O–C and C-O groups [23]. As for the previous region, the calculated spectra (Figure 2b,d) indicate similar bands to gelatin or CNC, which vary in intensity proportionally with the variation of the concentration of the components. However, this is not the case for the experimental spectra (Figure 2a,c), where a large number of bands vary in intensity and position.

The amide I region from gelatin structure has been extensively used to probe the protein structure and dynamics due to its sensitivity to the backbone conformation of a protein. It can also provide useful information about protein folding, misfolding and unfolding, as this spectral region is composed of overlapping bands arising from α-helices, β-sheets and nonordered structures. This vibrational mode comes from the stretching vibration of C=O from the peptide group, which is weakly coupled with the C-N stretching and in-plane N-H bending vibration [29,30]. The spectrum of pure gelatin presents two small bands at 1683 and 1668 cm^−1^ (visible more like shoulders than individual bands) and a strong band at 1626 cm^−1^, which are shifted to higher (1691, 1628 cm^−1^) and lower (1652 cm^−1^) wavenumber values in the composites’ spectra. Moreover, the band from 1691 cm^−1^ becomes visible as a band in the composites with 10% and 15% CNC, while the one from 1652 cm^−1^ appears like a strongly connected shoulder to the band from 1628 cm^−1^.

The shifting of the bands to higher/lower wavenumber is observed only in the experimental spectra, being an indicator that the reinforcing agent (CNC) influenced the gelatin structure and conformation.

In experimental spectra, the band corresponding to amide II from 1519 cm^−1^ (in G) presents a small shoulder at 1542 cm^−1^ (assigned to C-O stretching vibration) in the GC5 composite spectrum, which is shifted to 1548 cm^−1^ in CG10 and CG15 composite spectra and appears as a well-defined band with a small shoulder at 1521 cm^−1^. On the other side, in the calculated spectra this band remains unchanged (at 1518 cm^−1^). The amide II band is assigned to N-H bending and CN stretching vibrations. The modification of this band may indicate the changes in the hydrogen bonds near the peptide chains, therefore this modification might be due to the formation of new hydrogen bonds between O-H groups of CNC and N-H groups of proteins which partially weakens and perturbs the internal hydrogen bond network in gelatin [31].

Further modifications can be observed for the band from 1395 cm^−1^ (in G) (assigned to stretching vibration of C-N bonds [32]), which in the calculated spectra remain unchanged but in the experimental spectra is shifted to 1400 cm^−1^. In the spectra of the composites with 10% and 15% CNC, in their composition a new band at 1378 cm^−1^ can be identified. This later band it is assigned to CH deformation vibration in CNC and is shifted to a higher wavenumber compared to the pure component.

Other important modifications observed in the experimental spectra comparing to the calculated ones are in the 1130–1000 cm^−1^ region. Thus, the band from 1116 cm^−1^ (in G)/1107 cm^−1^ (in CNC) assigned to C-O-C stretching vibration in both components is shifted to 1111 cm^−1^ in the composite spectra; the band from 1081 cm^−1^ (in G) assigned to C-O stretching in gelatin decreases in the composite spectra, being observable only as a shoulder in GC15 composite, and is shifted to 1086 cm^−1^. In the calculated spectra, this band decreases in intensity proportionally with the increase of the CNC content. The bands from 1059 cm^−1^ (observable as a shoulder in G)/1028 cm^−1^ and 1055/1026 cm^−1^ (in CNC) assigned to C-O stretching vibration in both components are shifted to 1053 and 1032 cm^−1^, respectively. The first band appears as a well-defined band, while the second band presents a strong increase in intensity in the GC10 and GC15 composite spectra. Again, these bands’ intensity varies proportionally with the CNC content in the calculated spectra, while is not the case in the experimental ones.

Furthermore, the band from 920 cm^−1^ assigned to OH bending vibration increased in intensity in the experimental spectra of the composites but presented almost the same intensity in the calculated spectra. Clear differences by variation of band position and intensity in the experimental spectra comparing to calculated ones were observed, indicating the presence of chemical interactions taking place between the C-O, C-OH and N-H groups of the two components via hydrogen bond interactions.

#### 2.1.2. X-ray Diffraction (XRD)

X-ray diffraction analysis allows us to examine the microstructure of the polymeric composite films. Having an in-depth perception of the microstructure is very important when studying new materials, since it influences their physical and mechanical properties. The obtained diffraction patterns are depicted in Figure 3.

Gelatin films present a broad signal at 2θ of 20° assigned to the high content of amorphous faction and to the distance between adjacent polypeptide strands of G [33] and a small band at 7.7° assigned to triple-helix crystalline structures [32]. CNC diffractogram presented typical signals of cellulose I, such as: 15.2° signal assigned to the (1–10) plane, 16.4° signal assigned to (110) plane, 20.3° signal corresponding to (102) plane and 22.6° signal assigned to the (200) crystallographic plane [23,34]. The signals from 16.4° 2θ and 20.3° 2θ merged with the ones from 15.3° 2θ and 22.6° 2θ, being almost unnoticeable [35]. In the composite films, with the addition of CNC in gelatin matrix, the signals from 2θ = 15.3° and 22.6° increase in intensity. This can be correlated with the increasing of overall crystallinity in blends compared with gelatin film and can be a consequence of the formation of strong hydrogen bonds between the components [25]. Moreover, the signals from 15.3° and 16.4° are more visible in the diffractograms of the composite films and are shifted to higher values (to 15.5° and to 16.7°), indicating that interactions took place between the cellulose nanocrystals and gelatin matrix. These can be due to a rearrangement in the orientation of CNC crystallites after incorporation in the polymeric matrix [36]. Furthermore, with the addition of CNC, the band from 7.7° assigned to triple-helix crystalline structures from gelatin and the broad reflection from 20° decrease in intensity, being an indicator that the triple-helical structure is denaturated and the CNC interacts with the semicrystalline arrangement of the gelatin matrix [37].

#### 2.1.3. Scanning Electron Microscopy (SEM)

The surface morphology of gelatin-based nanocomposite films was evaluated by SEM (Figure 4).

Gelatin film exhibited a smooth and continuous surface due to the homogenous network structure. Adding CNC into the polymeric matrix led to a less smooth surface, indicating a microstructural change in the polymeric matrix, but the distribution of CNC is homogenous. However, the presence of white dots may be due to the appearance of some small cellulose nanocrystals aggregates and cross-sections of the nanocrystals added in the polymer matrix. These observations were also reported in the literature [12]. Li et al. [38] observed the formation of aggregation or inhomogeneous dispersion of CNC in the CMC matrix at higher than 7% content of the former one. Similar results were obtained by Mondragon et al. [39] when adding CNF into a gelatin matrix and by Popescu et al. [22] when CNC was added into κ-carrageenan nanocomposite films.

#### 2.1.4. Water Vapor Sorption Measurements

An important property of the coating materials is their behavior in contact with water molecules. Gelatin film exhibits a poor water vapor barrier property due to its hydrophilic characteristics of being a hygroscopic material, absorbing water depending on the RH at which it was stored [40]. The water sorption process implies the progressive penetration of water molecules through the polymeric matrix by chemisorption, physical adsorption and multilayer condensation. Generally, according to Van der Wel and Adan [41], the adsorption isotherms can be divided into three regions: a first region assigned to the single water molecules that do not show important interactions with other water molecules or with their environment; a second region assigned to water molecules which interact with other water molecules and form clusters or aggregates but are not big enough to have bulk water properties; and third region which is specific to the interaction of polymeric groups with the water molecules [42].

The adsorption isotherms of the gelatin and composite films represented as the moisture content as a function of relative humidity are shown in Figure 5.

As mentioned by Van der Wel, at lower values of relative humidity, the samples adsorb a small amount of water, and up to 45% RH all the samples exhibit almost the same behavior. After this RH value, the neat gelatin film starts to be more sensitive to water molecules compared to nanocomposite films. Over 70% RH, the sorption isotherms start to have an asymptotic increase. At 100% RH, the maximum MC value decreases with the increase of CNC content in the composite films, with neat gelatin film having about 163% MC, GC5—136%, GC10—128% and GC15—110%. The later one has a maximum MC 53% lower than in case of gelatin film.

This behavior may be a consequence of dispersion of nanofiller in the polymeric matrix, the presence of CNC in gelatin matrix causing a tortuous path for the water vapor molecules to enter through the films, and as observed by FTIR spectroscopy, intermolecular interaction between both polymers takes place due to the formation of hydrogen bonding throughout the polymer chain. Moreover, because of good interaction between the polymers, a more compact structure of film matrix results. This might be able to control and restrict the adsorption of water molecules. Ooi et al. [43] mentioned in their study that the reduction of water vapor sorption might be due to the increase in the crystallinity of the samples due to the introduction of CNCs or by reducing the available sorption sites (the free OH and NH hydrophilic groups) in the gelatin matrix, while George and Siddaramaiah [44] attributed the reduction of water vapor sorption ability to low hygroscopicity of highly crystalline CNCs.

To highlight the presence of water molecules in the composite films, NIR spectroscopy was used. The band from about 1940 nm is highly sensitive to the presence of water molecules being assigned to their stretching vibration. Figure 6a shows the integral area of this band in the case of GC15 composite film. The integral area increases with the increase of the MC in the films and the moisture content shows a linear dependence (R^2^: 0.99) when it is plotted as a function of the integral area of this band (Figure 6b).

#### 2.1.5. Moisture Adsorption Test

Generally, gelatin, CNC and glycerol are hydrophilic polymers and therefore can bind with water molecules, forming new hydrogen bonds [45]. Moreover, the porous structure of gelatin network can contribute to the easy absorbance of water molecules [46]. The value of moisture absorption of samples can be an indicator of water content within the composite films.

By exposing the films at 57% RH for 24 h, the moisture adsorption remained almost constant after 6 h.

The measured maximum moisture adsorption for gelatin film was 11.5% after 24 h (Figure 7a). The values decrease when CNC is added in the composite films. The lowest value for moisture adsorption was observed in case of the GC15 (6.8%) sample, being about 39% lower than in the case of gelatin film. In the case of nanocomposite films, due to hydrogen bonding interactions which took place between gelatin and CNC, the water uptake may be reduced because of reduction of chances of water molecules to bond to sorption sites.

Furthermore, Figure 7b plots the bands corresponding to adsorbed water in the films at 57% RH. The integral area of these bands is 5.9 (for G), 2.82 (for GC5), 2.42 (for GC10) and 1.67 (for GC15), showing a decrease with the increasing CNC content. These results agree with the results presented above.

The moisture content at equilibrium depends on the morphology (macro-voids, crystal size and crystallinity degree) and hydrophilicity of the nanocomposite films. Similar results were obtained by Syahida et al. [45] when adding palm wax into a fish gelatin matrix. Furthermore, Santos et al. [47] observed a decreased water transmission through gelatin-cellulose whisker films, which could be attributed to the reduction of amorphous areas and to a dense composite structure caused by the hydrogen-bonded percolating network of cellulose whiskers. Miao et al. mentioned in their study that “the uniform dispersion of the cellulose nanocrystals in the nanocomposite films would likely help block the permeating path of small molecules, and could thus contribute to the high barrier performance of the final nanocomposite film” [48].

#### 2.1.6. Determination of Swelling Degree

For swelling degree determination, the samples were immersed in distilled water. The gelatin film was completely dissolved after 5 min, but with an increase of the reinforcing agent, the resistance of samples increased (Figure 8).

The GC5 film after 10 min reached a swelling degree of almost 500% and after that the sample was dissolved. The other two composite films did not dissolve in water and reached a swelling degree of about 480% (GC10) and approximately 320% (GC15), respectively. As observed by FTIR spectroscopy, due to interactions taking place between the polymer matrix and reinforcement agent, a more compact structure of film was created. The increased resistance of the films can be attributed to the formation of intermolecular hydrogen bonds between hydroxyl groups of CNC and amino groups of gelatin chains, reducing their ability to interact with water molecules. Maroufi et al. [49] found in their study that the addition of modified carrageenan in gelatin reduced the solubility and moisture content of the films. They mentioned that this is due to the formation of crosslinking bonds between the amino groups from gelatin and the dialdehyde from modified carrageenan, leading to a denser structure with reduced water absorption ability and water binding capacity. In another study, Xiao et al. [50], using soy protein, CNC and zinc oxide nanoparticles, observed that the water solubility of the nanocomposite films was significantly reduced following the addition of CNC-containing agents. They mentioned that this reduction was due to “the formation of hydrogen bonds between the hydrophilic groups of protein and the surface hydroxyl groups of CNC which improved the cohesiveness and compactness of the film matrix, and thus decreasing the water sensitivity”.

#### 2.1.7. Water Contact Angle Measurements

Figure 9 shows the contact angle values of the studied films. All films, including the pure gelatin film, had contact angle values higher than 90°, which indicates that all the surface samples were slightly hydrophobic.

G film exhibited the lowest value for water contact angle (116°), which indicates it has more hydrophilic behavior compared to composite films. This is due to the fact that the polar amino acids in the gelatin backbone are more susceptible to moisture absorption [11]. Adding CNCs to gelatin increased the water contact angle value to 123° for GC5, to 124° for GC10 and to 127° for GC15 film. This could be due to formation of hydrogen bonds between gelatin polar groups and CNC hydroxyl groups [12], but also to a higher roughness due to CNCs dispersion within the gelatin matrix. Ahammed et al. [32] observed that WCA was 104.8° for gelatin film and the surface roughness induced by addition of zein in the films can significantly affect WCA values, decreasing it up to 50°. Moreover, Pereira et al. [37] found that the gelatin-based films exhibited values of WCA between 90° and 98°, with these values being influenced by the incorporation of ZnO, indicating that the increase of nanoparticle concentration enhanced the WCA of the films. Corrêa de Souza Coelho et al. [51] observed that adding CNCs into the starch matrix led to an increase of the WCA compared to starch control film, but this improvement was dependent on the content of the CNCs in the matrix. Therefore, for the incorporation of an amount up to 10% of CNCs the WCA increased, with this “behaviour corresponding to highly crystalline, hydrophobic characteristics of CNCs”. When the authors used a higher concentration of CNCs (15%), they observed similar values of the WCA with the control film. They mentioned that this can be due to a “possible aggregation caused by the higher CNC concentration in the matrix”. However, this behavior was not observed in our case, where the addition of 15% CNCs into the polymeric matrix induced a further increase in the contact angle.

#### 2.1.8. Dry-In Time of Water Droplets on the Surfaces and Absorption

The dry-in time of a water droplet is dependent on the relative humidity of the surrounding environment, being closely related to the contact angle and adsorption behaviors. If the ability of adsorption is high and the CA is low, the water droplets dry faster.

The upper side of Figure 10 shows the water droplets on the films immediately after dropping them and the lower side images show the water droplets after drying.

With the exception of gelatin film where the droplets were quickly absorbed, it was observed that the droplets remained on the surface of the nanocomposite films, presenting a low degree of absorption, and the dry-in time was of 30–35 min. This is due to the increasing resistance of the films reinforced with CNC compared to the neat gelatin film, where water droplets caused its dissolution. Another study [22] observed that the droplets remained on the surface of the carrageenan CNC films, presenting a low degree of absorption and having a dry-in time of about 40–45 min.

### 2.2. Evaluation of Seed Coating Influence on Seed Germination

#### 2.2.1. Compatibility Test

To be used as carriers for biofertilizers, the polymer matrices have to be compatible with them. It is known that the success of a bioformulation depends both on the microbial load and its survival mechanism, as well as on the compatible carriers used for the development of the formulation. Therefore, apart from a formulations’ storage stability, a non-compatible carrier can unfavorably affect the efficacy of the biocontrol agent.

In the present study, the compatibility between G and GC mixtures with *T. harzianum* KUEN 1585 (T) was determined.

From Figure 11, it can be observed that the polymeric coating did not inhibit the growth of *T. harzianum* KUEN 1585; moreover, the fungi developed on all the culture mediums, with higher colonial density towards the periphery of the petri dish in all cases. Therefore, both G and nanocomposite GCs were proven to be efficient carrier sources, as well as a nutrient source to the bio-agent.

Cortés-Rojas et al. [52] evaluated the *T. koningiopsis* spore compatibility with different polymers. They observed that alginate and potato starch showed the highest germination values for *T. koningiopsis* and the spores survival rate is related to the protection that polymers provide to the drying process and the moisture retention capacity during storage.

#### 2.2.2. Viability of *T. harzianum* KUEN 1585 on Corn Seeds Test

Usually, gelatin and pectin are used as water and air barrier films to extend the shelf life of fruits and vegetables, therefore they are considered good candidates for the microbial survival of the seeds by creating an environment that protects them from water vapor or reactive oxygen agents. Aside from a protecting role, these polymers are a good source of nutrients both for the microbes and the developing plants [52].

*T. harzianum*, in their natural environment, colonizes plant roots without apparent adverse reactions [48,49]. From Figure 12 it can be seen that after seven days the spores are viable on the seeds surface, because they improved the development of the roots. Seeds treated just with *T. harzianum* spores developed shorter roots (around 2 cm long), while seeds covered with gelatin developed roots with lengths between 5 cm (G + T) and 8 cm (GC10 + T) (7 cm—GC15 + T and 6 cm—GC5 + T). Therefore, when the fungal spores were added into the nanocomposite matrix, they led to an increase of the seed’s roots compared to simple fungal spores. As mentioned before, the nanocomposite matrix acted also as a nutrient for the fungi and plant. The best performance was identified for the GC10 + T nanocomposite formulation.

Our findings agree with the study of Swaminathan et al. [53]. They inoculated fungal spores on different polymeric formulations and observed that the survival time and the number of spores adhering to formulations was higher in case of xanthan gum, but in extreme conditions, like 30 °C and 79% RH, the best spore survival was observed for the starch formulation. Even though they did not use gelatin or CNCs in their study, we observed improved survival time and development of the plant roots which were similar, therefore, both gelatin and CNCs are great candidates for this type of formulation.

#### 2.2.3. Germination Test

To germinate, a seed must be kept in an environment that favors its development. The conditions that must be fulfilled are the existence of an adequate amount of water, temperature range and light (in some cases) [53]. To assess the impact of seed treatment and nanocomposite coating on the corn seeds, the germination percentage (as a measure of the extent to which the seeds have germinated), speed of germination (as a measure of the speed when the germination process has ended) and root length were measured. In Figure 13 are presented the seeds placed in germination chamber and in 7th day of germination. 

The seed quality parameters such as germination, speed of germination and root length were significantly influenced by the seed treatment, polymer coating and biofertilizer. *T. harzianum* is a free-living fungus which is common in root ecosystems and in soil, has been widely used as plant-growth enhancers, and also has an antagonistic effect against several pests [54,55]. López-Bucio et al. [56] suggested that *T. harzianum* acts under a mechanism of phytostimulation which involves a “multilevel communication with shoot and root systems”, promoting root branching and nutrient uptake capacity and liberating active metabolites into the rhizosphere (auxins, small peptides, volatiles), thereby stimulating plant growth and yield.

The germination percentage of the seeds coated only with polymeric composite compared with seeds coated with *T. harzianum* KUEN 1585 are presented in Figure 14. It can be seen that the germination percentage is improved when seeds are coated with the polymeric material. The lowest G% value is observed for control group (62%) and seeds treated only with G and *T. harzianum* KUEN 1585, respectively. Increasing the CNC content in the seed coating agent led to an increase in the G% up to 92% for GC10 and 90% for GC15. This can be attributed to the more compact structure of the coating that remains closer to the seed for a longer time, enhancing the germinability [20]. Furthermore, when the biofertilizer was added in the seed-coating material, the germination percentage improved, especially for gelatin and GC5 (Figure 14a). This fact can be assigned to the faster decomposition of gelatin, which enhanced the development of the biofertilizer, thus improving the G%. Moreover, in the case of seeds treated just with *T. harzianum* KUEN 1585, the G% was not improved, being an indicator that the development of the biofertilizer is influenced by the presence of the coating, which maintains the spores close to the seeds and may be acting as a nutrient for *Trichoderma* spp. after decomposition [20]. At the same time, the speed of germination was not significantly influenced by the polymeric coating, but when *T. harzianum* KUEN 1585 was added in the coating material the speed of germination improved, from 8.7 in the case of the control group up to almost 30 in the case of the GC15 + T group. This is an indicator that the germination process is accelerated due to the action of the biofertilizer [13]. Again, in this case, the values for speed of germination were higher when *T. harzianum* KUEN 1585 was incorporated in the coating material compared with the seeds treated just with the spores, indicating that the efficacy of the biofertilizer is improved when it is incorporated into the polymeric matrix [57].

On the 7th day, the length of roots was measured, and it can be seen they are correlated with the speed of germination. Coating the seeds with the polymeric material led to formation of longer roots. In consequence, the shorter roots were measured for uncoated seeds (3.38 cm) and when increasing the reinforcing agent, the values of the root lengths increased, reaching 13.2 cm for the GC15 group. Moreover, the addition of the biofertilizer improved this parameter, with the roots being longer in all cases compared with the seeds treated only with the polymeric coating. The longest roots were measured for the seeds treated with GC15 with *T. harzianum* KUEN 1585, having an average of 18.3 cm compared with the control lot, where the average of roots length was only 3.4 cm. The efficacy of *Trichoderma* spp. as a biofertilizer is given by the increasing of the nutrients’ solubility and nutrient uptake capacity of the roots [58] and also their homogenous distribution in the plants [59,60]. Kipngeno et al. [15] treated tomato seeds with *Bacillus subtilis* and *Trichoderma asperellum* as a seed coating for management of damping-off in tomatoes and found that the coating was a good growth promoter.

## 3. Materials and Methods

### 3.1. Materials

Gelatin (G) from bovine skin (Type B) and glycerol (N99.5%, MW: 92.02 g/mol) was purchased from Sigma Aldrich (St. Louis, MO, USA). Cellulose nanocrystals (CNCs) (sulfur content: 91 mmol*kg^−1^, Zeta-potential: −28.5 mV) was produced from cellulose pulp by sulfuric acid hydrolysis and kindly supplied by Melodea Ltd. (Rehovot, Israel). *Trichoderma harzianum* KUEN 1585 spores (Trademark Sim Derma) were provided by ORBA Biokimya (Istanbul, Turkey). Double distilled water was used for the solubilization of gelatin and as a solvent.

### 3.2. Film Preparation

A solution of 3 wt% gelatin (G) (prepared by the solubilization of the solid particles at 60 °C and 1500 rpm for 1 h in distilled water) was mixed with cellulose nanocrystals (CNCs) suspension 1.5 wt% in different ratios to reach the final concentrations of 0%, 5%, 10% and 15% CNC dry weight. Glycerol was used as a plasticizer and was added in the final solution (30% based on dry weight of components). The final mixture was homogenized by using magnetic stirring for 30 min at 60 °C and 1500 rpm and then mixed for 5 min with an ultraturax at 10,000 rpm to ensure a complete homogenization of the components in the blended solution. The resulting solutions were poured in Petri dishes and then dried at 45 °C for 24 h. The thickness of the films was of 0.15 ± 0.015 mm. The codes and the composition of nanocomposite films are presented in Table 1.

### 3.3. Film Characterization

#### 3.3.1. Fourier Transform Infrared Spectroscopy

The ATR-FTIR spectra were recorded at 4 cm^−1^ resolution in the 4000–500 cm^−1^ region on a Bruker ALPHA FTIR spectrometer using a diamond crystal. Five recordings were performed for each sample and the evaluations were made based on the average spectrum.

#### 3.3.2. X-ray Diffraction

The diffractograms were recorded on a Diffractometer D8 ADVANCE (Bruker AXS, Germany), using the CuKα radiation. The working conditions were 40 kV, 30 mA, 2 s/step, and 0.02°/step. All diffractograms were recorded in the 10–40 2θ degrees range at room temperature.

#### 3.3.3. Scanning Electron Microscope

The composite films were investigated using a scanning electron microscope (S4800 field emission SEM, Hitachi, UK). Each film was attached to SEM aluminum stub before being gold-coated for 90 s using a sputter coater (EMITECH K550X, Quorumtech, UK). During imaging, 3 kV and 8.5 mm were used as the acceleration voltage and observation distance.

#### 3.3.4. Water Vapor Sorption Measurements

Before testing, the samples were maintained in an oven for 24 h at 45 °C to reach the constant mass. Further, the samples were placed in desiccators with different RH values (3.5%, 8.5%, 16%, 25%, 36%, 45%, 54%, 59%, 69%, 75%, 80%, 85%, 92% and 100%), obtained by saturated salt solutions, at 25 ± 1 °C in a conditioned room. To record the exact values, a LogTag thermohygrometer was placed in each desiccator. Each sample was maintained at a certain RH until it reached a constant mass, then was weighed and transferred to the next container with a higher RH. The moisture content was calculated using the Equation (1) [42]:(1)$$\mathrm{MC}\;\%=\left[\left(\mathrm{Mf}-\mathrm{Mi}\right)\right]÷\mathrm{Mi}\times 100$$
where: MC is equilibrium moisture content, Mf is final mass and Mi is initial (dry) mass of the samples.

To identify the interactions taking place between the water and films, the NIR spectra were obtained using a PHAZIR Handheld Near-Infrared Analyzer (Thermo Fisher Scientific—Portable Optical Analysis, Waltham, MA, USA) in the range 1600–2400 nm, resolution 11 nm, scan number: 6.

#### 3.3.5. Moisture Adsorption Test

Dried samples were placed in a desiccator at 57% RH. The samples were weighed every hour until a constant mass was registered and the moisture content was calculated using the previous equation.

#### 3.3.6. Determination of Swelling Degree

Dried films were immersed in distilled water at 25 °C and weighted periodically at 5, 10, 15, 30, 45, 60, 90, 120, 180, 240 and 360 min (6 h). The water uptake (WU%) and percentage of absorbed water (M%) were calculated by using the same type of Equation (2) [42]:(2)$$\mathrm{WU}\;\%,\;M\;\%=\left[\left(\mathrm{Mf}-\mathrm{Mi}\right)\right]÷\mathrm{Mi}\times 100$$
where Mf is the final mass and Mi is the initial (dry) mass of the sample.

#### 3.3.7. Contact Angle Measurements

The static contact angles were determined by the sessile drop method, at room temperature and controlled humidity, within 30 s (the time corresponding to a metastable equilibrium between the liquid droplet and the tested surfaces) using a CAM-200 instrument from KSV, Finland. The measurements were performed using 1 μL drop of water on the film surface. Contact angle measurements were taken on five different locations on the surface and the average values were further considered. All measurements were done on the side of the films in contact with the air during drying.

#### 3.3.8. Dry-In Time of Water Droplets on the Surfaces and Absorption

To evaluate the dry-in time of a water droplet on the surface of the film samples, three droplets with comparable size were placed with a syringe on the surface of the films. To make the experiment more visible, the water was mixed with red beet extract (15% concentration) and the dry-in time of the droplets on the films surface was observed. By identifying the absorption of the droplets into the surface of the samples, the removability of the dried droplets was evaluated.

### 3.4. Seed Coating

In total, 2.5 mL from previous described solutions were added to 50 g corn seeds and mixed for 5 min. For seeds treated with the biocontrol agent, *Trichoderma harzianum* KUEN 1585 was added by 6 × 10^5^ CFU/10 g seeds into the initial polymeric solution. The covered seeds were dried at room temperature for 24 h.

#### 3.4.1. Compatibility Test

For the compatibility test, the poisoned food technique was used to evaluate the influence of the polymeric matrix on the development on *T. harzianum* spores [51]. For that, in the PDA medium (agar, 15 g/L, NaCl, 5 g/L, tryptone, 10 g/L, yeast extract, 5 g/L) 1.5 wt% solutions of G, GC5, GC10 and GC15 with a concentration of 1% were added. On the obtained medium, *T. harzianum* KUEN 1585 was placed, and the Petri dishes were kept at 25 °C for 7 days. Colony morphology and mycelial growth were observed daily. A Petri dish with PDA only with *T. harzianum* KUEN 1585 was used as a control. The influence of polymeric solutions on the development of *T. harzianum* KUEN 1585 was observed after 7 days. Twice the experiments were carried out in a randomized fashion in three replications for each treatment.

#### 3.4.2. Viability Test

To observe if the *T. harzianum* KUEN 1585 spores are still viable on the corn seeds, for each treatment one seed was placed in a Petri dish with PDA medium. After 7 days at 25 °C, the development of the *T. harzianum* KUEN 1585 spores was evaluated [61].

#### 3.4.3. Germination Test

The seeds were placed at 25 °C, 85% RH, in the dark for 7 days in paper towels for assessing the germination and seed quality parameters. Daily germination counts were recorded until no further germination occurred [62]. The number of normal seedlings and the length of the roots were counted after the test period and the germination (%) was calculated using the formula:Germination (%) = Number of normal seedlings/Total number of seeds × 100(3)

Speed of germination was calculated by the following formula [63]:Speed of germination = N1/T1 + N2/T2 + N3/T3 + … + Nx/Tx(4)
where N is the number of seeds germinated in days “T”.

#### 3.4.4. Statistical Analysis

A total of 100 seeds were used for each method: seed germination, speed of germination and root length determinations. Data were analyzed by analysis of variance. Means were separated by Fisher’s least significant difference (LSD) and *p* values < 0.05 were considered significant.

## 4. Conclusions

In this work, a new seed coating material using natural polymers was synthetized and characterized. By adding CNC as a reinforcing agent in gelatin film a more compact structure was obtained due to the chemical interactions established between the film components, mainly hydrogen bonds, as observed by FTIR and XRD. The good compatibility and homogenous distribution of CNC in the polymeric matrix demonstrated by SEM caused a tortuous path for the water vapor molecules to enter through the films, causing an improvement of resistance of the films with water molecules. Furthermore, to enhance the parameters that characterize seed germinability, *Trichoderma* spp. were immobilized in the seed coating matrix. It was observed that the biocontrol agent is compatible with the components of the nanocomposite films and remains viable on the seeds after coating. Moreover, treating the seeds with the biocontrol agent led to an improvement of G% (from 62% in case of control group up to 96% in case of GC5 with *T. harzianum*) and speed of germination. The length of the roots increased in the case of seeds treated with *T. harzianum* compared with seeds treated just with polymeric material. The results showed that immobilization of the biocontrol agent in the coating material of corn seeds has a promising potential for use in the agricultural sector.

## Figures and Tables

**Figure 1 molecules-26-05755-f001:**
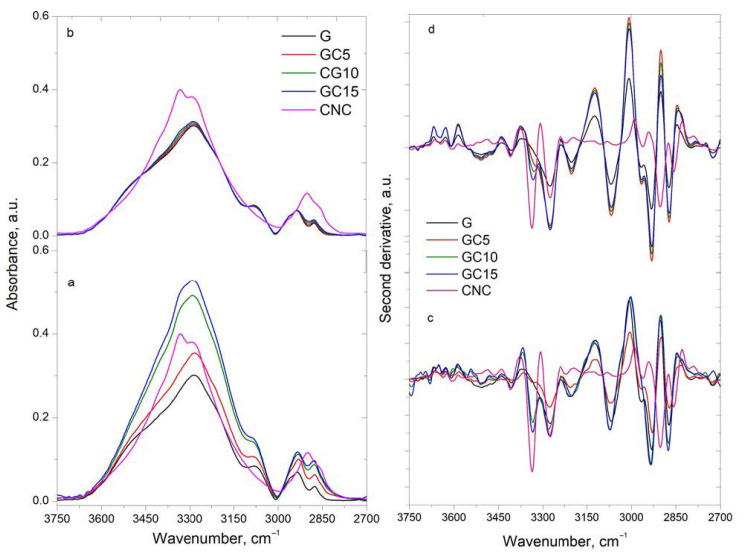
Experimental and calculated FTIR spectra (**a**,**b**) and their derivatives (**c**,**d**) in the 3750–2700 cm^−1^ region.

**Figure 2 molecules-26-05755-f002:**
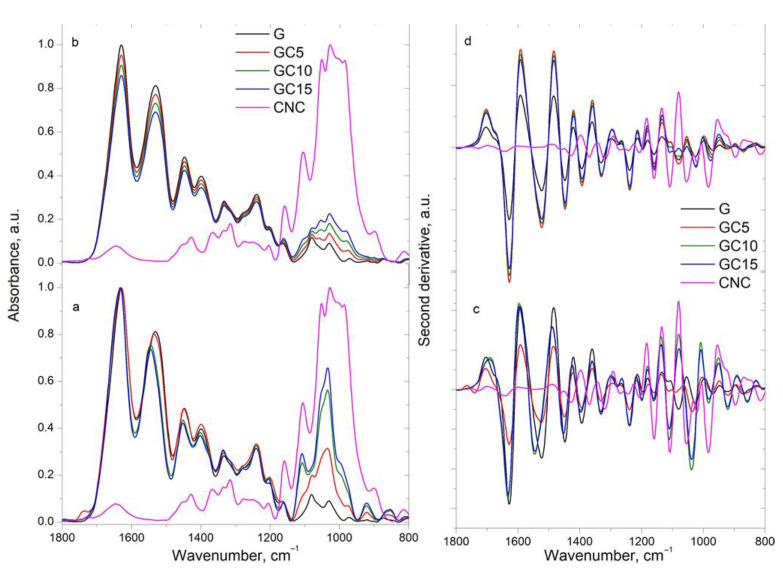
Experimental and calculated FTIR spectra (**a**,**b**) and their derivatives (**c**,**d**) in the 1800–800 cm^−1^ region.

**Figure 3 molecules-26-05755-f003:**
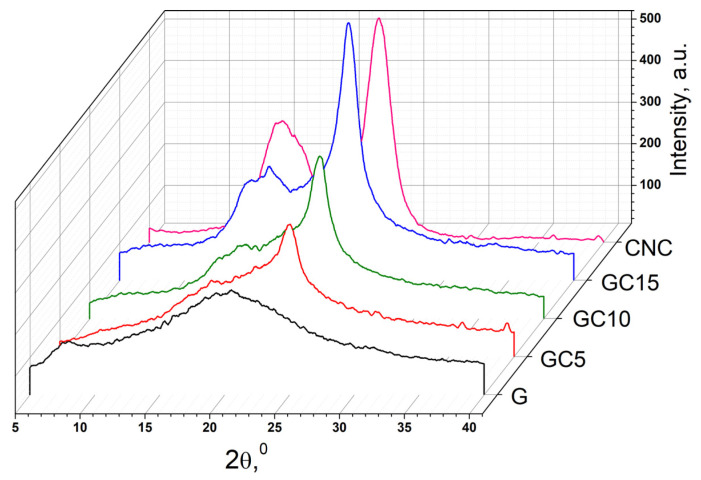
X-ray diffraction (XRD) patterns of gelatin, CNC and gelatin-based nanocomposite films.

**Figure 4 molecules-26-05755-f004:**
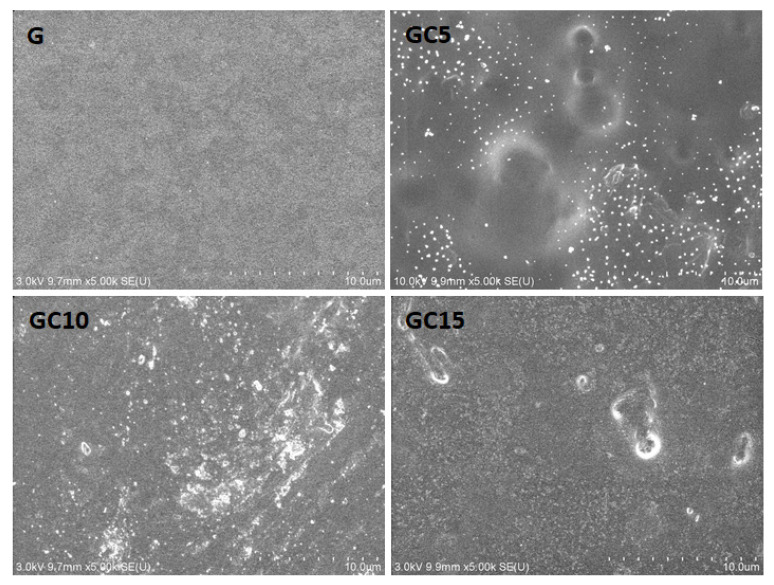
Morphological aspect of gelatin films reinforced with CNC.

**Figure 5 molecules-26-05755-f005:**
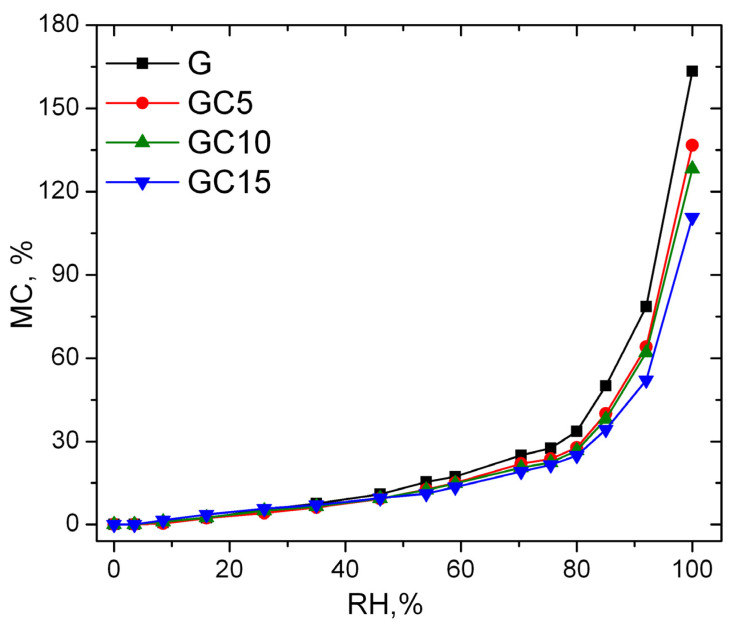
Sorption isotherms of the samples as a function of RH.

**Figure 6 molecules-26-05755-f006:**
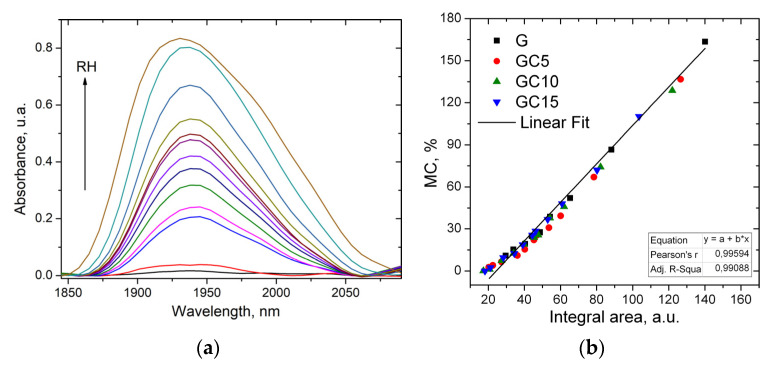
The band at 1940 nm for the GC15 sample in the sorption process (**a**), and the integral area plotted as a function of the MC for gelatin and composite films (**b**).

**Figure 7 molecules-26-05755-f007:**
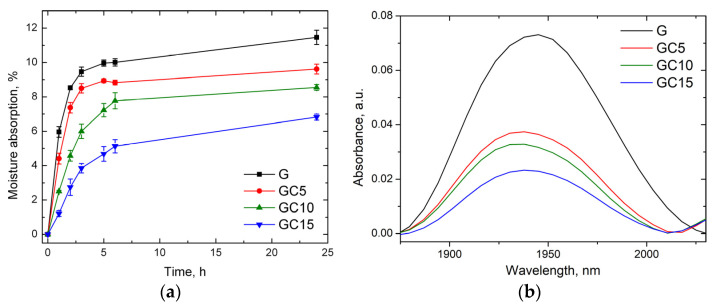
Moisture adsorption after exposing the gelatin and composite films at 57% for 24 h (**a**) and the NIR band from 1940 nm at 57% RH after 24 h of exposure (**b**).

**Figure 8 molecules-26-05755-f008:**
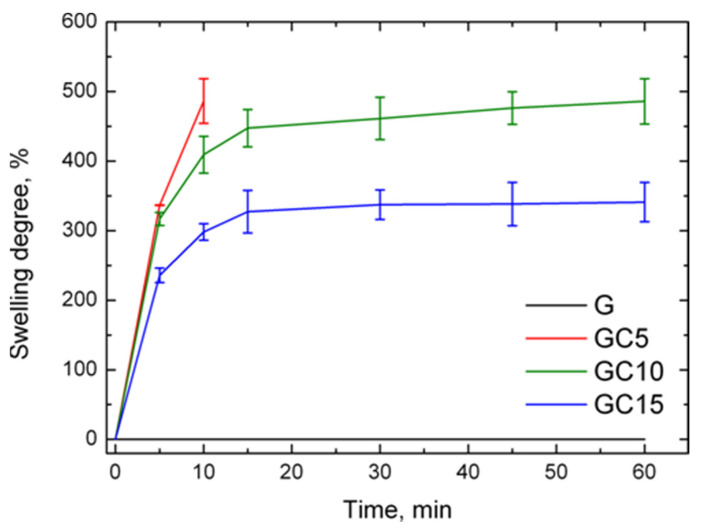
Swelling degree of gelatin and composite films immersed in water for 60 min.

**Figure 9 molecules-26-05755-f009:**
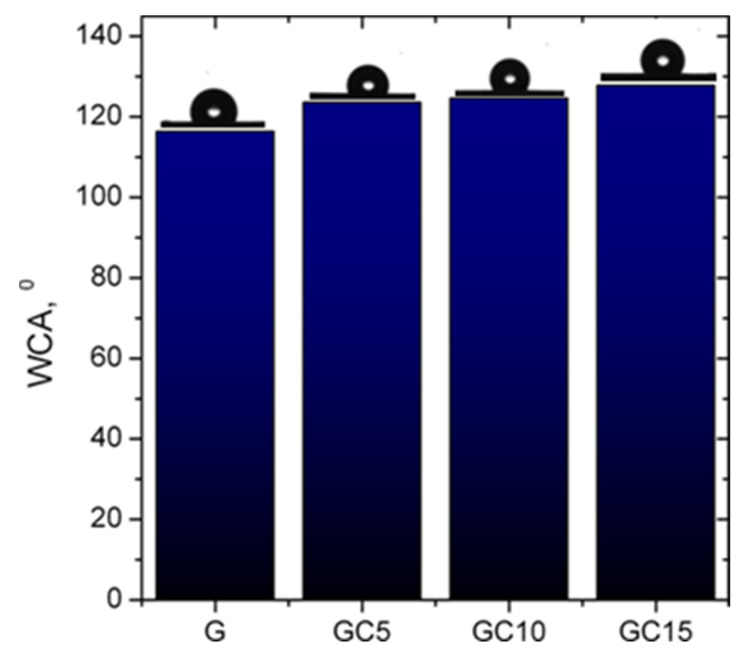
Values for water contact angles of the studied films and the images of the droplets on the surfaces of the samples.

**Figure 10 molecules-26-05755-f010:**
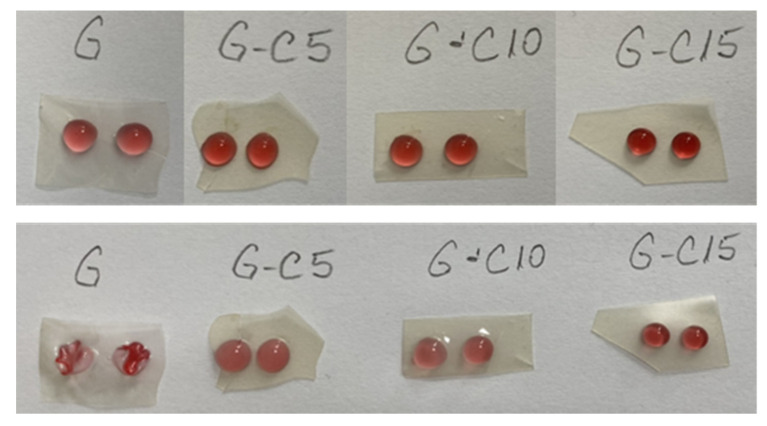
The water droplets on the surface of the films.

**Figure 11 molecules-26-05755-f011:**
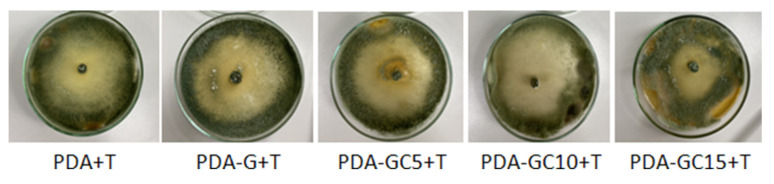
Growth of *T. harzianum* KUEN 1585 on PDA medium poisoned with G and GC mixtures on 7th day.

**Figure 12 molecules-26-05755-f012:**
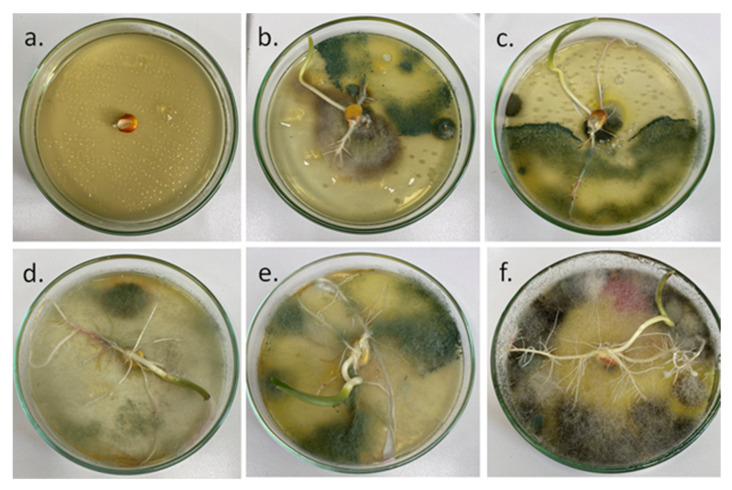
Viability test: corn seed on the first day (**a**), seeds on the 7th day (**b**–**f**): seeds treated with *T. harzianum* KUEN 1585 (**b**), G + T (**c**) GC5 + T (**d**), GC10 + T (**e**), GC15 + T (**f**).

**Figure 13 molecules-26-05755-f013:**
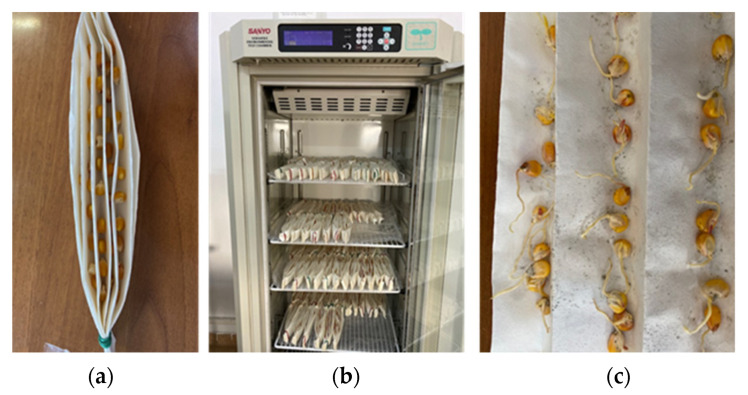
Seeds in paper towel for germination test (**a**), seeds placed in the germination chamber (**b**) and germinated seeds on 7th day (**c**).

**Figure 14 molecules-26-05755-f014:**
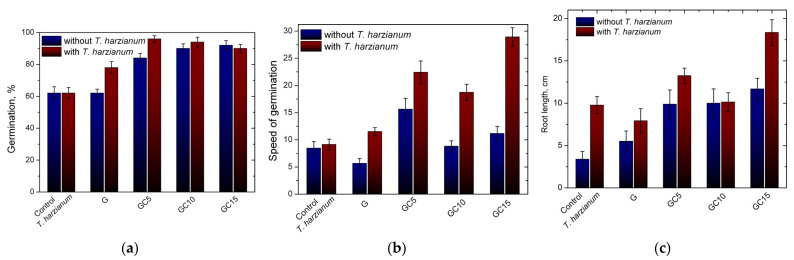
Germination percentage (**a**) speed of germination (**b**) and length of roots (**c**) of coated seeds with and without *T. harzianum* KUEN 1585 after 7 days of germination.

**Table 1 molecules-26-05755-t001:** Sample codes and compositions.

Code	G, %	CNC, %
G	100	-
GC5	95	5
GC10	90	10
GC15	85	15

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
