# Peer review of "Gelatin Reinforced with CNCs as Nanocomposite Matrix for Trichoderma harzianum KUEN 1585 Spores in Seed Coatings"

_molecules, 2021, doi:10.3390/molecules26195755_

Round 1

Reviewer 1 Report

In this paper, authors introduced a new material to be applied in agriculture. The characteristics of this novel materials have the different properties. However, authors did not have enough information in agriculture. Please ask the help of n agronomist to revise this paper.

  1. Figure 5. Sorption isotherms of the samples as a function of RH.

The sorption isotherms are affected by ambient temperature. Please describe the temperature effect. That is, provide the sorption isotherms perperties in different temperatures.

  1. Line 301-303 As can be observed, the integral area increases with the increase of the MC in the films and the moisture content shows a linear dependence when it is plotted as a function of the integral area of this band (Figure 6b).

Please use the regression analysis to evaluate the linear or nonlinear. Not to use the “be observed”.

  1. Figure 7. Moisture adsorption after exposing the gelatin and composite films at 57% for 24h (a) and the NIR band from 1940 nm at 57% RH after 24h of exposure (b).

Same as Figure 5, what is the effect of temperature?

4.Line4. 322-324: Furthermore, from the Figure 7b it can be seen that the band corresponding to adsorbed water is decreasing with the increasing of CNC content in the films at 57% RH, this fact being in correlation with the results presented above.

  Please use the quantitative method, not to use the “it can be seen”. That is an subjective way.

  1. About the 2.2.2. Viability of T. harzianum KUEN 1585 on corn seeds test and 2.2.3. Germination test

 What are the rules or guides about these tests? Please list the sources.

  1. Figure 14. Germination percentage (a) speed of germination (b) and length of roots (c) of coated seeds with and without T. 438harzianum KUEN 1585 after 7 days of germination.

 Please use the statistical test for comparison.

Reviewer 2 Report

General comments

The submitted paper reports on the production and characterisation of gelatin film reinforced with cellulose nanocrystals to be used as seed coating material.

The topic well matches the aim and scope of Molecules. However, as reported below in details, the Results and Discussion section has to be improved. Moreover, the Authors should better evidence the originality of their work and the added value to the scientific knowledge about the considered topic.

A deep English grammar and language revision is strongly recommended

More details and specific remarks and suggestions are reported below point by point.

Abstract

  • The acronyms, such as FT-IR, XRD, SEM, have to be written in the extended version the first time they are use.
  • Please correctly rewrite the following phrase “By water vapour sorption, swelling degree and contact angle measurements was observed an improvement in water vapour properties of the films.”

Keywords

The chosen keywords (i.e. gelatin; CNC; seed coating; biofertilizer) do not cover the manuscript content. Please add further ones. Moreover, CNC has to be replaced with the extended name.

Introduction

- The following statements “Pesticides have been used in agriculture for increasing the food productivity since World War II, having positive impact on pest management” and “However, despite all the advantages, the use of pesticides is avoided nowadays due to the social and environmental damage, but also due to their negative effect on the human health” have to be supported with recent appropriate references, including “Reusable optical multi-plate sensing system for pesticide detection by using electrospun membranes as smart support for acetylcholinesterase immobilisation, Mater Sci and Eng C 111 (2020), 110744.”

- The following sentence “One of the most studied polysaccharide-based nanomaterials in polymer nano-composites are cellulose nanocrystals (CNC). Generally, CNCs are obtained by acid hydrolysis, degrading the amorphous regions of the cellulose fibers and preserving the  crystalline ones”, as well as “The main properties, which make CNCs to be largely used, are their biodegradability, renewability, non-toxicity and abundance.” needs to be corroborated with appropriate references, including “Effect of silver nanoparticles and cellulose nanocrystals on electrospun poly(lactic) acid mats: morphology, thermal properties and mechanical behaviour, Carbohydrate Polymers 103 (2014): 22– 31”.

  • The Authors should better evidence the originality of their work and the added value to the scientific knowledge about the considered topic.

  1. Results and Discussions

This section should be expanded and improved, sicne more quantitative data have to be reports for some characterisations, as reported below. Many data and figures have to be not only described, but also discussed. Some results have to be justified and compared with the literature data.

2.1. Film Characterization

2.1.1. Fourier Transform Infrared Spectroscopy

- All the FTIR peaks assignments have to be supported with proper references.

- The following sentence “For an immiscible blend, the calculated and the experimental spectra should be the same, while for a miscible blend, differences can occur due to the interactions between the components” needs to be corroborated with suitable references, including “Electrospun PHBV/PEO co-solution blends: microstructure, thermal and mechanical properties, Materials Science and Engineering: C 33[3] (2013): 1067–1077”.

2.1.3. Scanning electron microscopy (SEM)

- The Authors stated that “The SEM images show the presence of a good compatibility between the components due to the high number of hydrogen bonds established among the two components.”, but form the reported SEM images it is not possible to evidence a good compatibility. In order to highlight a good wettability , the sections should be shown. Moreover, for the presence of the hydrogen bonds among the two components, they should refer to the related analysis (FTIR) and spectra.

2.1.7. Water contact angle measurements

- The following consideration “This is due to the fact that the polar amino acids in the gelatin backbone are more susceptible for moisture absorption” has to be supported with suitable references.

- Higher water contact angle values in the case of composite films can be ascribed not only to the formation of hydrogen bonds between gelatin polar groups and CNC hydroxyl groups, but also to the higher roughness due to the CNCs dispersion within the gelatin matrix (loto effect).

2.1.8. Dry-in time of water droplets on the surfaces and absorption

- The results have not only to be described, but also discussed and justified.

- The following conclusion “For gelatin film, the water droplets caused the dissolution of the film.” has to be explained and justified, also supporting it with suitable references.

2.2. Evaluation of seed coating influence on seed germination

2.2.1. Compatibility test

The reported images have to be better described and discussed. Moreover some quantitative data have to be added.

2.2.2. Viability of T. harzianum KUEN 1585 on corn seeds test

The reported images have to be better described and discussed. Moreover some quantitative data have to be added.

2.2.3. Germination test

- The following considerations “This can be attributed to the more compact structure of the coating that remained a longer time near to the seed, enhancing the germinability”, “This fact can be assigned to the faster decomposition of gelatin which enhanced the development of the biofertilizer, thus improving the G%”, “Moreover, in the case of seeds treated just with T. harzianum KUEN 1585, the G% was not improved, being an indicator that the development of the biofertilizer is influenced by the presence of the coating, which maintain the spores close to the seeds and maybe acting as a nutrient for Trichoderma spp. after the decomposition” and  “This is an indicator that the germination process is accelerated due to the action of the biofertilizer” have to be supported with proper references.

  1. Materials and Methods

As a general consideration, for all the reported equations and relations suitable references have to be added.

3.1. Materials

- For gelatin and glycerol more details have to be added, such as the molecular weight.

- For CNCs, the synthesis procedure has to be reported in details.

3.2. Film preparation

- For the gelatin solution it has to be specified that water was used as solvent.

- For CNCs it is not correct to talk of solution but it is a suspension.

3.3.4. Water vapor sorption measurements

- For NIR analysis, the resolution and scans number have to be specified.

3.3.8. Dry-in time of water droplets on the surfaces and absorption

The proportion between water and red beet extract has to be reported.

3.4.1. Compatibility test

- The composition of the poisoned food technique has to be specified.

- The composition of PDA medium has to be reported.

- The concentration of the seeded T. harzianum KUEN 1585 has to be specified.

- How was the colony morphology investigated?

- How was the mycelial growth quantitatively evaluated?

3.4.2. Viability test

- How was the development of the T. harzianum KUEN 1585 spores quantified?

Round 2

Reviewer 1 Report

The content of revised manuscript have been improved significantly. It could be accepted.

Reviewer 2 Report

The paper can be accepted in the current version.